# Spatial Decoupling Method for a Novel Dual-Orthogonal Induction MEMS Three-Dimensional Electric Field Sensor

**DOI:** 10.3390/mi16040381

**Published:** 2025-03-27

**Authors:** Jiacheng Li, Junpeng Wang, Chunrong Peng, Wenjie Liu, Jiahao Luo, Zhengwei Wu, Ren Ren, Yao Lv

**Affiliations:** 1State Key Laboratory of Transducer Technology, Aerospace Information Research Institute, Chinese Academy of Sciences, Beijing 100190, China; lijiacheng221@mails.ucas.ac.cn (J.L.); wangjunpeng22@mails.ucas.ac.cn (J.W.); liuwenjie23@mails.ucas.ac.cn (W.L.); luojiahao23@mails.ucas.ac.cn (J.L.); zwwu@mail.ie.ac.cn (Z.W.); renren@aircas.ac.cn (R.R.); lvyao@aircas.ac.cn (Y.L.); 2University of Chinese Academy of Sciences, Beijing 100049, China

**Keywords:** 3D electric field, MEMS electric field sensor, decoupled calibration matrix, coupling interference, dual-orthogonal induction, Cramér–Rao lower bound

## Abstract

To mitigate the three-dimensional (3D) coupling interference of electric field sensors, a novel MEMS 3D electric field sensor with a dual-orthogonal induction structure and its spatial decoupling method is proposed. The sensor is designed with a cylindrical structure, in which two pairs of induction electrodes are orthogonally arranged to suppress common-mode interference. MEMS electric field sensing chips are utilized to achieve 3D electric field measurement. Furthermore, a spatial decoupling calibration model is established based on the structural characteristics of the sensor. The Cramér–Rao lower bound of the linear model is calculated to obtain the optimal decoupled calibration matrix, enabling precise 3D electric field decoupling. Experimental results showed that within an electric field range of 0–50 kV/m, the linearity of the three decoupled electric field components was 2.60%, 1.20%, and 1.78%, respectively, while the synthesized electric field achieved a linearity of 0.74% with a maximum full-scale error of 0.80%. Under varying angles and field intensities, the maximum and average relative errors of the decoupled synthesized electric field were 1.20% and 0.43%, respectively, representing reductions of 61.8% and 56.1% compared to the conventional matrix inversion method. These results confirmed that the proposed method effectively suppressed coupling interference and enhanced 3D electric field measurement accuracy.

## 1. Introduction

Atmospheric 3D electric field detection technology plays a crucial role in various fields, including atmospheric science, space physics, aviation safety, and meteorological disaster warning [1,2,3,4]. For instance, in atmospheric science, it enables the analysis of charge distribution within thunderclouds and their evolution mechanisms, facilitating lightning prediction [5]. In space physics, electric field measurements contribute to studying geomagnetic storms and plasma fluctuations, which are essential for space environment monitoring and satellite communication security [6]. In aviation safety, real-time electric field information is used to assess the impact of atmospheric electric field disturbances on aircraft, ensuring the safety of aerospace missions [7]. Due to the complexity of the atmospheric electric field, which is influenced by the charge distribution within clouds and exhibits a 3D dynamic non-uniform distribution [8], it is necessary to employ electric field sensors for accurate 3D field measurements.

Currently, 3D electric field sensors for aerial atmosphere primarily include airborne, field-mill, and micro-electro-mechanical system (MEMS) types [9,10,11]. Airborne electric field sensors are typically mounted on aircraft, allowing for 3D field measurements by adjusting the aircraft’s attitude. However, their measurement accuracy is often affected by the stringent calibration requirements and interference from aircraft-induced charging effects [12]. Field-mill electric field sensors detect electric field signals by periodically exposing and shielding electrodes, thereby enhancing their anti-interference capability. Nevertheless, their complex mechanical structure and susceptibility to wear over prolonged operations can compromise measurement stability [13,14]. In contrast, MEMS 3D electric field sensors have attracted significant attention in recent years for their compact size, low power consumption, and high integration, making them a focal point in airborne electric field measurement research [11].

However, in practical measurements, the output signal of the sensing electrodes is influenced not only by the electric field component perpendicular to the electrode surface but also by interference from the remaining two directional components caused by inter-axis coupling. This coupling effect significantly degrades the accuracy of electric field measurements. To minimize such interference, various optimization strategies have been proposed. For example, Ling et al. [15] designed a MEMS 3D electric field sensor with low inter-axis coupling, employing a symmetric structure and differential circuits to suppress electric field coupling interference along non-measurement axes, achieving an inter-axis coupling sensitivity better than 3.48%. Similarly, Peng et al. [16] introduced a 3D electric field sensor chip featuring a single-shielded electrode, utilizing differential detection to enable 3D electric field measurements with a maximum measurement error of less than 5.3%. While these methods have effectively mitigated inter-axis coupling interference through structural optimization at the chip level, they still face limitations in achieving high-precision atmospheric 3D electric field detection, as inter-axis coupling is not completely eliminated, making it challenging to fully meet the requirements for accurate 3D field measurements.

To further enhance electric field measurement accuracy and mitigate coupling interference, researchers have proposed a series of decoupling calibration methods. Mach et al. [12] developed a decoupling approach for distributed airborne field models by establishing a relationship matrix between sensor output signals and actual electric field components based on the inverse of the sensitivity coefficient matrix. Although this method enables 3D field decoupling, its calibration process is complex and requires high-precision instrumentation. To improve decoupling accuracy while simplifying the calibration procedure, Li et al. [17] proposed a decoupling calibration method based on a genetic algorithm, which optimizes the fitness function and genetic operators to obtain an optimal decoupled calibration matrix. This approach eliminates errors associated with traditional matrix inversion, achieving a maximum relative error of only 1.90% for synthesized electric field measurements. Nevertheless, the overall calibration framework is not yet fully refined. Wu et al. [18] further improved the calibration model by incorporating dimensional coupling and angular deviation. Using a differential evolution algorithm, they solved the sensitivity coefficient matrix, effectively reducing system errors. However, this method does not fully account for coupling noise effects on calibration accuracy and remains susceptible to local optima. Zhao et al. [19] proposed a support vector regression-based decoupling calibration method, leveraging multi-objective optimization to derive the optimal decoupling matrix and minimize computational errors. While this approach offers notable advantages, its effectiveness is highly dependent on kernel function selection and hyperparameter tuning, resulting in increased computational complexity. Overall, existing 3D airborne electric field measurement techniques still face challenges in measurement accuracy, anti-interference capability, and decoupling calibration methodologies.

In recent years, with the rapid development of artificial neural networks and other technologies, some emerging decoupling methods such as Extreme Learning Machine (ELM) and Radial Basis Function (RBF) neural networks, have been widely applied in fields such as multi-dimensional force sensors and acceleration sensors [20,21,22]. These methods usually employ complex algorithms, such as the Whale Optimization Algorithm (WOA) with the ELM, or utilize RBF neural networks to achieve sensor decoupling. By learning the complex relationships in the sensor calibration data, these methods can effectively decouple sensors and improve measurement accuracy to some extent. However, these methods also have some limitations. First, the complexity of the algorithms limits the real-time performance of the decoupling process. In addition, some of these methods may have lower linearity, resulting in larger errors. In addition, neural network methods are prone to overfitting problems during training, which affects their generalization ability in practical applications, resulting in potential instability and reliability issues in real-world environments.

To address the challenge of inter-axis coupling interference in 3D electric field measurements using electric field sensors, this paper proposes a novel MEMS electric field sensor with a dual-orthogonal induction structure, along with a spatial electric field decoupling method. The sensor features two pairs of orthogonally arranged sensing electrodes on the surface of a cylindrical structure, effectively suppressing common-mode interference and enabling accurate 3D electric field measurements. Additionally, a spatially symmetric structure is employed to achieve differential decoupling. Building on this sensor architecture, a 3D electric field decoupling calibration model is established, and an optimized spatial decoupling method is introduced. By computing the Cramér–Rao lower bound (CRLB) under a linear model, the optimal decoupled calibration matrix is derived to enhance measurement accuracy. Furthermore, a coordinate transformation model for 3D electric field calibration is established, and a sensor prototype along with a dedicated calibration fixture is developed. A calibration and testing system is constructed to experimentally validate the decoupling performance of the proposed method. Experimental results demonstrate that the proposed method significantly reduces coupling interference between electric field components. The computed synthetic field closely aligns theoretical values, exhibiting smaller errors compared to the conventional sensitivity matrix inversion method. These findings confirm that the proposed decoupling approach effectively mitigates inter-axis coupling, reduces measurement errors, and improves both the accuracy and efficiency of 3D electric field measurement.

## 2. Dual-Orthogonal Induction MEMS 3D Electric Field Sensor

### 2.1. Sensor Structure Design and Operating Principle

The structure of the proposed MEMS 3D electric field sensor with a dual-orthogonal induction configuration is illustrated in Figure 1. The sensor consists of the sensing electrodes, sensing chips, internal signal processing circuits, and a housing. The sensing electrodes are arranged in a dual orthogonal differential configuration, with two pairs of electrodes aligned along the X- and Y-axes, while an additional height difference is introduced along the *Z*-axis to ensure measurement orthogonality in three dimensions.

According to Gauss’s theorem, the relationship between the three spatial electric field components $E_{x}$, $E_{y}$, and $E_{z}$ and the induced charges on the sensing electrodes can be expressed as follows:(1)$$E_{x}=\frac{Q_{2}-Q_{1}}{\varepsilon (S_{1}+S_{2})}$$(2)$$E_{y}=\frac{Q_{4}-Q_{3}}{\varepsilon (S_{3}+S_{4})}$$(3)$$E_{z}=\frac{Q_{1}+Q_{2}-(Q_{3}+Q_{4})}{\varepsilon (S_{1}+S_{2}+S_{3}+S_{4})}$$
where $Q_{1}$, $Q_{2}$, $Q_{3}$, and $Q_{4}$ are the induced charges on the four sensing electrodes when subjected to an electric field in the respective directions. $S_{1}$, $S_{2}$, $S_{3}$, and $S_{4}$ denote the effective sensing areas of the corresponding electrodes, and ε is the permittivity of the surrounding medium.

As observed from Equations (1)–(3), the symmetrical electrode pairs along the X- and Y-axes ensure that the induced electric field signals in each respective direction are equal in magnitude but opposite in polarity. This configuration enables differential amplification to achieve the decoupling of the lateral electric field components. Along the *Z*-axis, the height difference between the electrode pairs ensures that the induced electric field signals are also equal in magnitude but opposite in direction, thereby enhancing the measurement accuracy of the *Z*-axis electric field component.

The sensor measures the external electric field based on the principle of charge induction. The sensor’s sensing electrodes are electrically connected to the surface of the sensing chips via conductive traces. The working principle of the sensitive structure is illustrated in Figure 2. When the external electric field $E_{0}$ is applied to the sensing electrodes, a periodic excitation signal modulates the system, causing the grounded shielding electrodes to induce a cyclic shielding effect on the positive and negative sensing electrode combs. This modulation generates an alternating induced signal that is directly proportional to the intensity of the external electric field. The induced signal is then amplified and converted into a voltage signal through the internal signal processing circuit, thereby enabling the accurate measurement of the external electric field.

### 2.2. Basic Decoupling Method of Spatial 3D Electric Field Measurements

A spatial electric field can be decomposed into three orthogonal components in any Cartesian coordinate system. During 3D electric field measurements, the output voltage of the sensor is affected by coupling interference among these components, which degrades measurement accuracy. Assuming that the sensor output noise follows a zero-mean Gaussian distribution, the relationship between the output voltage of the sensing chip i in the MEMS 3D electric field sensor and the vector electric field can be expressed as(4)$$u_{i}-u_{i0}=k_{ix}E\prime_{x}+k_{iy}E\prime_{y}+k_{iz}E\prime_{z}+n_{i}$$
where $u_{i0}$ is the output voltage of the sensing chip i in the absence of an external electric field, and $u_{i}$ is its real-time output voltage. The terms $E\prime_{x}$, $E\prime_{y}$, and $E\prime_{z}$ represent the three orthogonal components of the measured electric field E in the sensor coordinate system. The parameters $k_{ix}$, $k_{iy}$, and $k_{iz}$ denote the sensitivity coefficients of sensing chip i to each electric field component, which depend on the sensor’s structural design and the arrangement of the sensing chips. The term $n_{i}$ represents the Gaussian noise of the sensor output signal, where $n_{i}\sim N(0,\sigma_{i}^{2})$.

For a MEMS 3D electric field sensor composed of multiple sensing chips, the relationship between the output voltages of all sensing chips and the electric field components can be expressed in matrix form as follows [20]:(5)$$\left[\begin{array}{c} u_{1}\\ u_{2}\\ \vdots \\ u_{m} \end{array}\right]-\left[\begin{array}{c} u_{10}\\ u_{2}0\\ \vdots \\ u_{m0} \end{array}\right]=\left[\begin{array}{ccc} \begin{array}{c} k_{1x}\\ k_{2x}\\ \vdots \\ k_{mx} \end{array} & \begin{array}{c} k_{1y}\\ k_{2y}\\ \vdots \\ k_{my} \end{array} & \begin{array}{c} k_{1z}\\ k_{2z}\\ \vdots \\ k_{mz} \end{array} \end{array}\right]\left[\begin{array}{c} E\prime_{x}\\ E\prime_{y}\\ E\prime_{z} \end{array}\right]+\left[\begin{array}{c} n_{1}\\ n_{2}\\ \vdots \\ n_{m} \end{array}\right]$$
where m represents the number of sensing chips (m ≥ 3). The first term on the left-hand side is the real-time output voltage vector of the sensing chips, denoted as U, while the second term is the output voltage vector under zero-field conditions, denoted as $U_{0}$. On the right-hand side, the first term is the sensitivity coefficient matrix, denoted as K; the second term represents the 3D electric field components in the sensor coordinate system, denoted as $E^{\prime }$; the third term is the noise vector, denoted as N. Theoretically, the sensitivity coefficient matrix K uniquely determines the mapping relationship between the output voltage and the measured electric field. Equation (5) can be rewritten in vector form as follows:(6)$$U-U_{0}=KE^{\prime }+N$$

In practical measurements, the actual electric field components need to be decoupled from the output voltage of the 3D electric field sensor. By constructing a decoupled calibration matrix, the estimated value of the measured electric field $\hat{E^{\prime }}$ can be expressed as(7)$$\hat{E^{\prime }}=C(U-U_{0})$$
where C is the decoupled calibration matrix. By appropriately calculating the matrix C, the coupling interference among electric field components can be effectively compensated, thereby enabling high-precision 3D electric field measurements.

## 3. Spatial Decoupling Method for 3D Electric Field Measurement Based on the CRLB

### 3.1. Decoupling Model for the MEMS 3D Electric Field Sensor

Based on Equations (6) and (7), the accurate measurement of the target electric field requires sensor decoupling calibration using standard electric fields. Assuming that n sets of calibration data are available, the relationship between the output voltage $U_{j}$ of the 3D electric field sensor and the input electric field $E_{j}$ in the *j*-th calibration set can be expressed as(8)$$U_{j}-U_{0}=K{E^{\prime }}_{j}+N_{j}$$
where $N_{j}$ represents the sensor noise in the *j*-th dataset, assumed to follow a zero-mean Gaussian distribution $N_{j}\sim N(0,\Sigma_{j})$, with $\Sigma_{j}$ being the covariance matrix of the sensor noise, given by $\Sigma_{j}=diag(\sigma_{1,j}^{2},\sigma_{2,j}^{2},\sigma_{3,j}^{2},\sigma_{4,j}^{2})$. To eliminate bias effects, the bias-corrected output voltage ${U^{\prime }}_{j}$ is introduced as ${U^{\prime }}_{j}=U_{j}-U_{0}$. To obtain the optimal sensitivity coefficient matrix K, the residual function between the observed and theoretical voltage values is minimized:(9)$$\underset{K}{\min}\overset{n}{\underset{j=1}{\sum }}{\left\|{U^{\prime }}_{j}-K{E^{\prime }}_{j}\right\|}_{\Sigma_{j}}^{2}$$

Since noise characteristics may vary among different sensing chips, a weighted linear least squares method is applied to optimize the residual function. The optimization objective function $J_{1}\left(K\right)$ is defined as(10)$$J_{1}\left(K\right)=\overset{n}{\underset{j=1}{\sum }}{\left({U^{\prime }}_{j}-K{E^{\prime }}_{j}\right)}^{T}W_{j}\left({U^{\prime }}_{j}-K{E^{\prime }}_{j}\right)$$
where $W_{j}$ is the noise weight matrix, defined as the inverse of the sensor noise covariance matrix: $W_{j}=\Sigma_{j}^{-1}$. Taking the derivative of $J_{1}\left(K\right)$ with respect to K, and equating it to zero, yields(11)$$\frac{\partial J_{1}\left(K\right)}{\partial K}=-2\overset{n}{\underset{j=1}{\sum }}W_{j}{U^{\prime }}_{j}{{E^{\prime }}_{j}}^{T}+2\overset{n}{\underset{j=1}{\sum }}W_{j}K{E^{\prime }}_{j}{{E^{\prime }}_{j}}^{T}=0$$

Rearranging the equation gives(12)$$\overset{n}{\underset{j=1}{\sum }}W_{j}{U^{\prime }}_{j}{{E^{\prime }}_{j}}^{T}=K\overset{n}{\underset{j=1}{\sum }}W_{j}{E^{\prime }}_{j}{{E^{\prime }}_{j}}^{T}$$

Thus, the sensitivity coefficient matrix K can be expressed as(13)$$K=\left(\overset{n}{\underset{j=1}{\sum }}W_{j}{U^{\prime }}_{j}{{E^{\prime }}_{j}}^{T}\right){\left(\overset{n}{\underset{j=1}{\sum }}W_{j}E\prime_{j}{{E^{\prime }}_{j}}^{T}\right)}^{-1}$$

During standard field calibration, if the covariance matrix of the sensor noise remains approximately constant across different calibration sets ($\Sigma_{1}=\Sigma_{2}=\cdots =\Sigma_{n}$), Equation (13) can be further simplified as(14)$$K=\left(\overset{n}{\underset{j=1}{\sum }}{U^{\prime }}_{j}{{E^{\prime }}_{j}}^{T}\right){\left(\overset{n}{\underset{j=1}{\sum }}E\prime_{j}{{E^{\prime }}_{j}}^{T}\right)}^{-1}$$

By defining $U^{\prime }=\left[{U^{\prime }}_{1},\ldots ,{U^{\prime }}_{n}\right]$, $E^{\prime }=\left[{E^{\prime }}_{1},\ldots ,{E^{\prime }}_{n}\right]$, the sensitivity coefficient matrix K can be expressed as(15)$$K=\left(U^{\prime }{E^{\prime }}^{T}\right){\left(E^{\prime }{E^{\prime }}^{T}\right)}^{-1}$$

During the calibration process, standard electric fields are applied sequentially in an orthogonal manner to ensure that $E^{\prime }{E^{\prime }}^{T}$ is invertible. Thus, Equation (15) uniquely determines the sensitivity coefficient matrix, enabling the high-precision decoupling calibration of the sensor.

### 3.2. Decoupling Method Based on the CRLB

After obtaining the sensitivity coefficient matrix K, it is necessary to further establish the relationship between the measured electric field and the sensor’s output voltage to achieve decoupling. According to the linear coupling relationship in Equation (6), the optimal estimation of the measured electric field $E^{\prime }$ can be determined by calculating the CRLB.

The CRLB is a theoretical lower bound used to quantify the estimation accuracy. It provides the minimum achievable variance for an unbiased estimator under ideal conditions. In simple terms, the CRLB gives the smallest possible estimation error, serving as an important tool for evaluating the performance of an estimator.

In this decoupling model, the CRLB is used to calculate the minimum achievable estimation error of the sensor output under a given model assumption. This lower bound allows us to assess the theoretical minimum error when using a specific decoupling method, thereby providing a theoretical basis for optimizing algorithms.

Suppose we want to estimate a vector $x=[x_{1},x_{2},\ldots ,x_{n}]$, and its measurement are obtained through a random process. The CRLB is defined based on the likelihood function $p\left(x\right)$, and the formula is as follows:(16)$$CRLB(x)={\left[E\left(\frac{\partial \ln p(x)}{\partial x}\right){\left(\frac{\partial \ln p(x)}{\partial x}\right)}^{T}\right]}^{-1}$$

In Equation (16), $p\left(x\right)$ is the likelihood function of the parameter x, and $\frac{\partial \ln p(x)}{\partial x}$ is the gradient of the log-likelihood function. In this coupling model, the sensor’s bias-corrected output voltage $U^{\prime }$ follows a Gaussian distribution, with its likelihood function $p(U^{\prime };E^{\prime })$ expressed as
(17)$$p\left(U^{\prime };E^{\prime }\right)=\frac{1}{4\pi^{2}{\left|\Sigma \right|}^{\frac{1}{2}}}\exp\left(-\frac{1}{2}{\left(U^{\prime }-KE^{\prime }\right)}^{T}\Sigma^{-1}\left(U^{\prime }-KE^{\prime }\right)\right)$$
taking the logarithm of the likelihood function and computing its second derivative with respect to $E^{\prime }$ yields(18)$$\frac{\partial^{2}\ln p\left(U^{\prime };E^{\prime }\right)}{\partial E^{\prime }\partial {E^{\prime }}^{T}}=-K^{T}WK$$
Thus, the Fisher information matrix $I\left(E^{\prime }\right)$ is obtained as(19)$$I\left(E^{\prime }\right)=-E\left[\frac{\partial^{2}\ln p\left(U^{\prime };E^{\prime }\right)}{\partial E^{\prime }\partial {E^{\prime }}^{T}}\right]=K^{T}WK$$

According to the CRLB theory, the covariance matrix of any estimator $\tilde{E^{\prime }}$ of the measured electric field satisfies the following inequality:(20)$$Cov\left[\tilde{E^{\prime }}\right]\geq I^{-1}\left(E^{\prime }\right)$$
where $I^{-1}\left(E^{\prime }\right)$ is the inverse of the Fisher information matrix. Thus, the CRLB can be expressed as(21)$$CRLB=I^{-1}\left(E^{\prime }\right)={\left(K^{T}WK\right)}^{-1}$$

From Equation (21), the optimal linear unbiased estimation of the measured electric field $E^{\prime }$ should be based on ${\left(K^{T}WK\right)}^{-1}$, which corresponds to the weighted least squares (WLS) estimation. According to the linear coupling relationship in Equation (6), the optimization objective function $J_{2}\left(E^{\prime }\right)$ for $E^{\prime }$ is given by(22)$$J_{2}\left(E^{\prime }\right)={\left(U^{\prime }-KE^{\prime }\right)}^{T}W\left(U^{\prime }-KE^{\prime }\right)$$
taking the derivative of $J_{2}\left(E^{\prime }\right)$ with respect to $E^{\prime }$ and setting the result to zero yields the optimal estimation $\hat{E^{\prime }}$ of the measured electric field:(23)$$\hat{E^{\prime }}={\left(K^{T}WK\right)}^{-1}K^{T}WU^{\prime }$$

Verify that $\hat{E^{\prime }}$ is the optimal linear estimator by computing its expectation:(24)$$E\left[\hat{E^{\prime }}\right]=E\left[{\left(K^{T}WK\right)}^{-1}K^{T}WU\prime \right]=E^{\prime }$$
It is confirmed from Equation (24) that it is an unbiased estimator. The covariance matrix is then calculated as(25)$$Cov\left[\hat{E^{\prime }}\right]=E\left[\left(\hat{E^{\prime }}-E^{\prime }\right){\left(\hat{E^{\prime }}-E^{\prime }\right)}^{T}\right]={\left(K^{T}WK\right)}^{-1}K^{T}W\Sigma W^{T}K{\left(K^{T}WK\right)}^{-1}={\left(K^{T}WK\right)}^{-1}$$
From Equation (25), it is evident that $\hat{E^{\prime }}$ attains the CRLB, implying that it achieves the minimum variance. Therefore, the estimator in Equation (23) is the optimal linear estimator.

Comparing Equation (23) with Equation (7), the decoupled calibration matrix C under optimal linear estimation can be expressed as(26)$$C={\left(K^{T}WK\right)}^{-1}K^{T}W$$

By calculating the decoupled calibration matrix C using Equation (26), the 3D electric field sensor can be accurately calibrated, effectively eliminating coupling interference among electric field components and enabling high-precision electric field measurements.

## 4. Coordinate Transformation Model and Calibration System

### 4.1. Calibration Coordinate Transformation Model Based on Rotation Matrix

To calibrate the 3D electric field sensor, it is essential to apply a known standard electric field to establish the relationship between the sensor’s output voltage and the applied field. However, in calibration experiments, the standard electric field is typically defined in a global coordinate system (GCS) referenced to the calibration setup, while the relationship between the sensor’s output voltage and the input field is established in the sensor’s local coordinate system (LCS). Therefore, to accurately express the components of the input electric field in the sensor’s coordinate frame, a coordinate transformation model must be established to convert the standard electric field from the GCS to the LCS.

Assume that the three orthogonal axes of the GCS are denoted as OX, OY, and OZ, while those of the LCS are denoted as $OX^{\prime }$, $OY^{\prime }$, and $OZ^{\prime }$, as illustrated in Figure 3. Initially, the LCS is aligned with the GCS, meaning their coordinate axes coincide completely.

During the calibration process, the sensor undergoes rotations around its local coordinate axes, causing the LCS to deviate from the GCS. Suppose the sensor rotates sequentially by angles α, β, and γ about $OZ^{\prime }$, $OY^{\prime }$, and $OX^{\prime }$ axes, respectively. The rotation directions follow the right-hand rule (i.e., counterclockwise rotation is considered positive when viewed from the positive semi-axis). Under these conditions, the standard electric field $E={\left[\begin{array}{ccc} E_{x} & E_{y} & E_{z} \end{array}\right]}^{T}$ in the GCS is transformed into its components in the LCS, denoted as $E^{\prime }={\left[\begin{array}{ccc} {E^{\prime }}_{x} & {E^{\prime }}_{y} & {E^{\prime }}_{z} \end{array}\right]}^{T}$, through a rotation matrix R, which is expressed as(27)$$E^{\prime }=RE$$
where R is the rotation matrix that describes the coordinate transformation from the GCS to the LCS. The matrix representation of this transformation is given by(28)$$\left[\begin{array}{c} {E^{\prime }}_{x}\\ {E^{\prime }}_{y}\\ {E^{\prime }}_{z} \end{array}\right]=\left[\begin{array}{ccc} \cos\alpha \cos\beta & -\sin\alpha \cos\beta & \sin\beta \\ \sin\alpha \cos\gamma +\cos\alpha \sin\beta \sin\gamma & \cos\alpha \cos\gamma -\sin\alpha \sin\beta \sin\gamma & -\cos\beta \sin\gamma \\ \sin\alpha \sin\gamma -\cos\alpha \sin\beta \cos\gamma & \cos\alpha \sin\gamma +\sin\alpha \sin\beta \cos\gamma & \cos\beta \cos\gamma \end{array}\right]\left[\begin{array}{c} E_{x}\\ E_{y}\\ E_{z} \end{array}\right]$$

This matrix describes the transformation of the standard electric field components from the GCS to their corresponding representations in the LCS after undergoing rotational changes. By utilizing this coordinate transformation model, the known standard electric field in the GCS can be accurately converted to the LCS, thereby establishing a precise mapping between the input electric field and the sensor’s output voltage. This facilitates sensor calibration and decoupling computations.

### 4.2. Sensor Prototype and Calibration System

In this study, a MEMS 3D electric field sensor prototype was developed, with its external casing and internal structure illustrated in Figure 4. The sensor features sensing chips as its core components, integrated with a meteorological data acquisition module, a wireless communication module, a data processing unit, and a battery-powered system. This configuration enables real-time electric field measurements in complex environments. The sensor housing is made of foamed plastic material to enhance thermal insulation. Its surface is covered with a layer of aluminum foil, which is electrically connected to the sensor’s ground to prevent charge accumulation. The housing measures 25 cm in length with an outer diameter of 5.6 cm. The surface-sensing electrodes are made of gold material, with both length and width measuring 3.5 cm. The sensing electrodes adopt a dual-orthogonal configuration, employing a differential detection method to enhance measurement accuracy and improve resistance to interference in 3D electric field detection. Additionally, a wireless communication module is utilized to transmit measurement data, eliminating potential electric field distortions caused by traditional wired connections and ensuring accuracy and stability during the calibration process. The sensing chip of the MEMS electric field sensor is designed and fabricated based on the SOI-MEMS process [23]. The physical prototype of the MEMS electric field sensor is shown in Figure 5.

To standardize the calibration procedure and ensure the accuracy and completeness of data acquisition in the decoupling calibration process, a 3D electric field decoupling calibration system was constructed, as schematically illustrated in Figure 6.

This system primarily consists of calibration plates, a pair of high-voltage DC power supplies, a data receiver, and a calibration fixture. The calibration plates are composed of parallel positive and negative electrodes powered by the high-voltage DC sources, generating a uniform electric field between them. The spacing between the plates is set to 20 cm, and the DC power supply can generate a high voltage ranging from 0 to 20 kV, ensuring a stable and controllable calibration environment. The sensor’s wireless communication module emits a signal at a frequency of 400 MHz, which is received by the data receiver. The received data are then transmitted to the host computer via the RS232 protocol, ensuring real-time monitoring while preventing external wiring from interfering with the standard electric field. Furthermore, a calibration fixture was designed to allow the multi-angle rotation of the sensor. The fixture consists of two fixed shafts and a central rotating ring. The sensor is mounted on the rotating ring and can rotate freely along the fixed shafts, enabling electric field calibration and measurement at various orientations.

### 4.3. Decoupling Calibration Procedure

The decoupling calibration procedure for the 3D electric field sensor is illustrated in Figure 7 and consists of the following steps.

(1) Initial Calibration: Mount the sensor on the calibration fixture such that the LCS coincides with the GCS ($R_{1}=I$). Apply a standard electric field ranging from 0 to 50 kV/m, calculate the electric field $E\prime_{1}$ in the LCS, and record the sensor output $U_{1}$.

(2) Rotation Calibration around the OZ′ axis: Rotate the sensor counterclockwise by 90° around the OZ′ axis (obtain $R_{2}$). Repeat the application of the standard electric field ranging from 0 to 50 kV/m, calculate the electric field $E\prime_{2}$ in the LCS, and record the second sensor output $U_{2}$.

(3) Rotation Calibration around the OX′ axis: Based on the previous step, rotate the sensor counterclockwise by 90° around the OX′ axis (obtain $R_{3}$). Repeat the application of the standard electric field ranging from 0 to 50 kV/m, calculate the electric field $E\prime_{3}$ in the LCS, and record the third sensor output $U_{3}$.

(4) Decoupling Calculation: Use the three sets of calibration data to calculate the sensitivity matrix K and the decoupled calibration matrix C, completing the sensor decoupling calibration.

This decoupling process ensures the uniqueness of the sensitivity matrix K and enables the accurate decoupling of the 3D electric field.

## 5. Experimental Validation and Results

### 5.1. Calibration Results

Following the calibration procedure, the triaxial calibration of the sensor was conducted within the calibration system, and the corresponding calibration curves are presented in Figure 8. As shown in Figure 8, during *X*-axis calibration, the differential configuration of the sensing electrodes ensures that the outputs of sensing electrodes 1 and 2 exhibit opposite polarity, achieving differential decoupling along the *X*-axis. Simultaneously, sensing electrodes 3 and 4 suppress coupling interference from the *X*-axis electric field through a differential structure. Similarly, during *Y*-axis calibration, sensing electrodes 3 and 4 output signals with opposite polarity, achieving differential decoupling along the *Y*-axis, while sensing electrodes 1 and 2 effectively suppress *Y*-axis coupling interference. For *Z*-axis calibration, the four sensing electrodes are mutually orthogonal with a height difference between the electrode pairs. Consequently, sensing electrodes 1 and 2 produce signals in the same direction, whereas sensing electrodes 3 and 4 generate signals with opposite polarity, enabling differential decoupling along the *Z*-axis.

Based on the acquired calibration data, two decoupling methods were employed to compute the calibration matrix: the CRLB-based optimization method and the sensitivity matrix inversion method. The resulting decoupling calibration matrices are as follows:

The optimized decoupled calibration matrix $C_{opt}$ based on the CRLB is(29)$$C_{opt}=\left[\begin{array}{cc} \begin{array}{cc} 12.3824 & -9.1911 \end{array} & \begin{array}{cc} 4.0723 & 4.8480 \end{array}\\ \begin{array}{cc} -0.2127 & 0.1589 \end{array} & \begin{array}{cc} -14.7831 & 15.2420 \end{array}\\ \begin{array}{cc} 5.3764 & 8.7850 \end{array} & \begin{array}{cc} -12.7045 & -12.6108 \end{array} \end{array}\right]$$

The decoupled calibration matrix $C_{mi}$ obtained via the sensitivity matrix inversion method is(30)$$C_{mi}=\left[\begin{array}{cc} \begin{array}{cc} 11.1543 & -11.2116 \end{array} & \begin{array}{cc} 2.6360 & 3.2646 \end{array}\\ \begin{array}{cc} -0.5705 & -0.4298 \end{array} & \begin{array}{cc} -15.2016 & 14.7906 \end{array}\\ \begin{array}{cc} 7.0518 & 11.5413 \end{array} & \begin{array}{cc} -10.7452 & -10.4509 \end{array} \end{array}\right]$$

### 5.2. Three-Dimensional Electric Field Testing

The performance of the sensor was evaluated. The sensor was rotated to a random orientation, and a standard electric field ranging from 0 to 50 kV/m was applied in 10 kV/m increments. The three components of the electric field and the resultant electric field were calculated using the decoupling method based on the CRLB, and their variations with respect to the applied electric field were plotted, as shown in Figure 9. The linearity of the decoupled electric field components in the X-, Y-, and Z-directions was calculated to be 2.60%, 1.20%, and 1.78%, respectively, while the linearity of the resultant electric field was 0.74%. The maximum relative error over the full measurement range was 0.80%.

Furthermore, the sensor was rotated to different orientations, and standard electric fields of 20 kV/m, 30 kV/m, 40 kV/m, and 50 kV/m were applied. The accuracy of the resultant electric field measurements obtained using the two decoupling methods was evaluated, as summarized in Table 1. The results indicate that for different sensor orientations and input electric field conditions, the calculated resultant electric field values obtained using both decoupling methods were in good agreement with the actual values, effectively eliminating coupling interference and enabling accurate 3D electric field measurements. In all 18 tests, 15 of the tests showed that the decoupled electric field’s relative errors obtained using the proposed method were significantly smaller than those obtained with the traditional methods, while the remaining 3 tests showed comparable errors between both methods. This demonstrates that the proposed method achieves higher accuracy under a variety of electric field conditions. Moreover, at Angle 4 and an electric field strength of 20 kV/m, the relative error of the traditional method was significantly higher than that of the proposed method. This suggests that at lower electric field strengths, the decoupling performance of the traditional method is limited and more prone to coupling interference, while the proposed method exhibits greater stability under these conditions.

Table 2 summarizes the relative errors between the calculated and theoretical resultant electric field values obtained using the two decoupling methods and provides a comparison with the best-performing decoupling method reported in [17]. As shown in Table 2, compared to the sensitivity matrix inversion-based decoupling method, the CRLB-based method reduced the maximum error from 3.14% to 1.20%, representing a 61.78% decrease. This demonstrates the ability of the proposed method to effectively suppress coupling interference among electric field components, thereby improving the accuracy of the electric field calculations. When compared to the genetic algorithm-based decoupling method presented in [17], the proposed method achieves a 36.84% reduction in the maximum error, showing its robustness in practical applications. Similarly, the average error decreased from 0.98% to 0.43%, a reduction of 56.12%. In comparison with the genetic algorithm-based method from [17], the proposed method achieved a 27.12% reduction in average error, further highlighting its superiority in practical measurements. In conclusion, the CRLB-based decoupling method proposed in this paper demonstrates a significant improvement in accuracy under various electric field conditions. Compared to the traditional sensitivity matrix inversion method and the genetic algorithm-based method in [17], the proposed method is more effective in suppressing coupling interference, leading to enhanced 3D electric field measurement accuracy. Therefore, the proposed method provides a more reliable solution for precise decoupling in electric field measurements, with broad potential for applications that require high accuracy.

## 6. Conclusions

In this paper, a dual-orthogonal induction MEMS electric field sensor and a spatial electric field decoupling method were proposed to mitigate coupling interference in 3D electric field measurements. The sensor employs two pairs of orthogonally arranged sensing electrodes on the surface of its cylindrical structure to suppress common-mode interference, while MEMS-based electric field-sensing chips enable precise 3D electric field measurements. A 3D electric field decoupling calibration model was established based on the structural characteristics of the sensor, and a linear optimal spatial decoupling method was developed. By deriving the CRLB for the linear model, an optimal linear estimation of unknown electric fields was achieved. Additionally, a coordinate transformation model for sensor calibration was constructed, and a sensor prototype along with a calibration fixture was designed.

A dedicated calibration and test system was built to experimentally validate the proposed decoupling method. Results demonstrated that within an applied electric field range of 0–50 kV/m, the proposed decoupling method resulted in linearities of 2.60%, 1.20%, and 1.78% for the X, Y, and Z components, respectively, while the resultant electric field exhibited a linearity of 0.74% with a maximum full-scale relative error of 0.80%. Compared to the sensitivity matrix inversion-based decoupling method, the CRLB-based approach reduced the maximum relative error of the resultant electric field from 3.14% to 1.20%, and the average relative error from 0.98% to 0.43%. These results indicate that the proposed MEMS-based electric field sensor and spatial decoupling method effectively suppressed coupling interference, significantly improving the accuracy of 3D electric field measurements.

Although the MEMS electric field sensor and decoupling method proposed in this paper have shown excellent performance in experiments, there are still some challenges in real-world applications. The long-term stability of the sensor may be affected by material fatigue and performance degradation of the MEMS components, thus requiring regular calibration or design improvements to ensure sustained accuracy. Additionally, variations in temperature and humidity can impact the sensor’s precision. Future research could focus on enhancing the sensor’s environmental adaptability, such as by designing temperature compensation algorithms or using more stable materials. Finally, future work could explore more effective decoupling methods and noise suppression techniques to enable 3D electric field detection in complex environments, thereby improving the system’s practicality and reliability.

## Figures and Tables

**Figure 1 micromachines-16-00381-f001:**
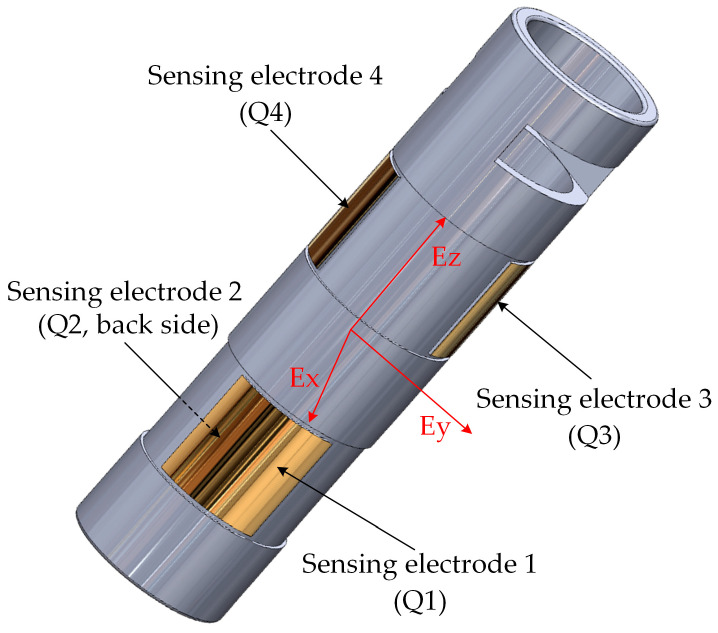
Structure of the dual-orthogonal inductive MEMS 3D electric field sensor.

**Figure 2 micromachines-16-00381-f002:**
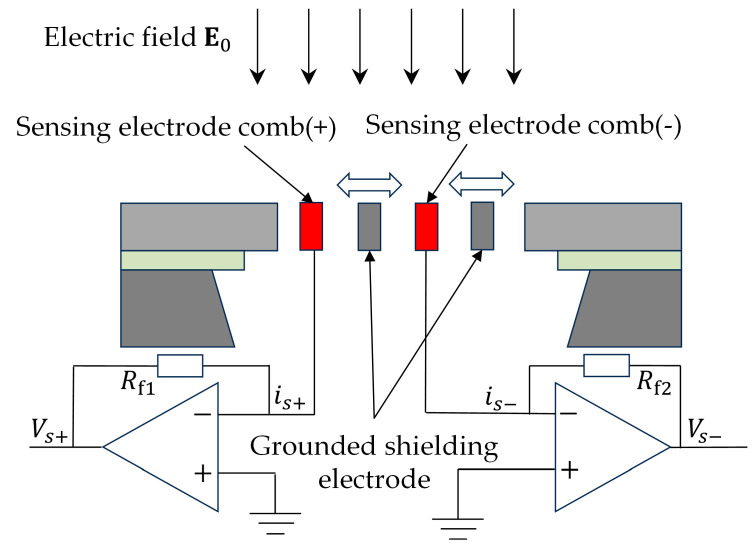
Working principle of the sensing structure.

**Figure 3 micromachines-16-00381-f003:**
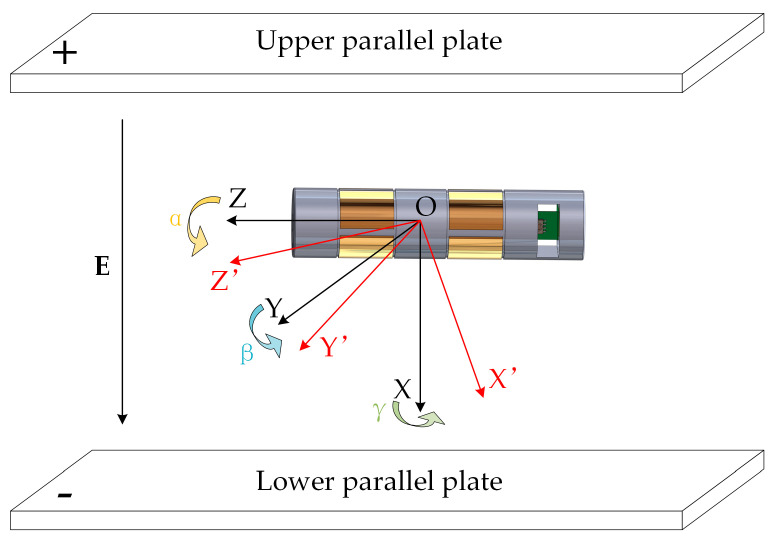
Coordinate system transformation model of the sensor.

**Figure 4 micromachines-16-00381-f004:**
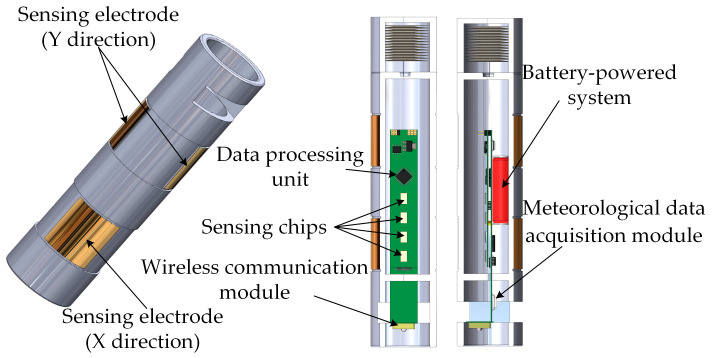
Sensor housing and internal structure.

**Figure 5 micromachines-16-00381-f005:**
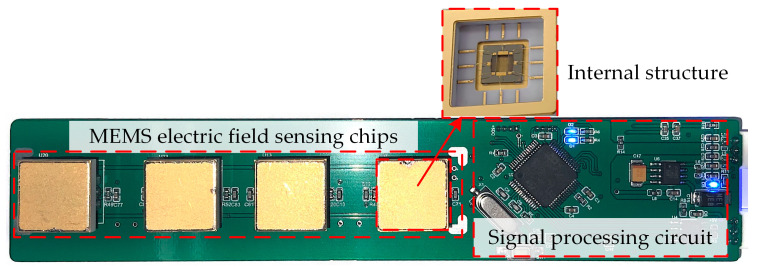
Physical prototype of the MEMS electric field sensor.

**Figure 6 micromachines-16-00381-f006:**
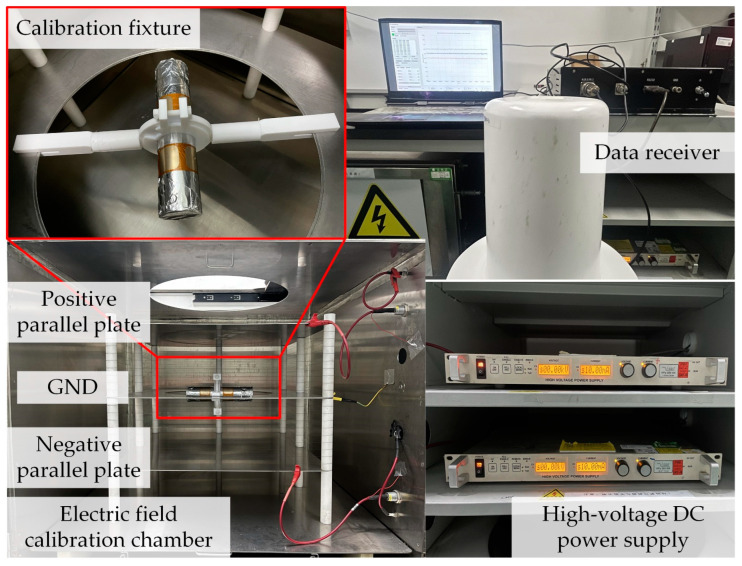
Three-dimensional electric field decoupling calibration system.

**Figure 7 micromachines-16-00381-f007:**
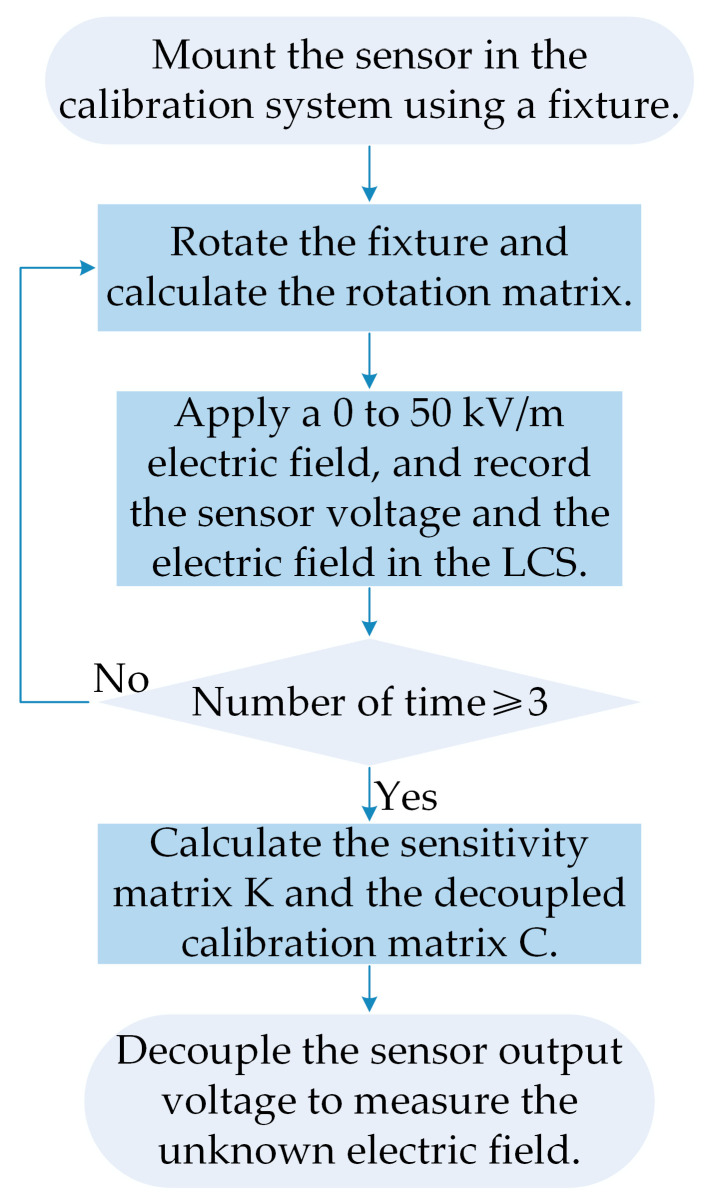
Decoupling calibration procedure for the 3D electric field.

**Figure 8 micromachines-16-00381-f008:**
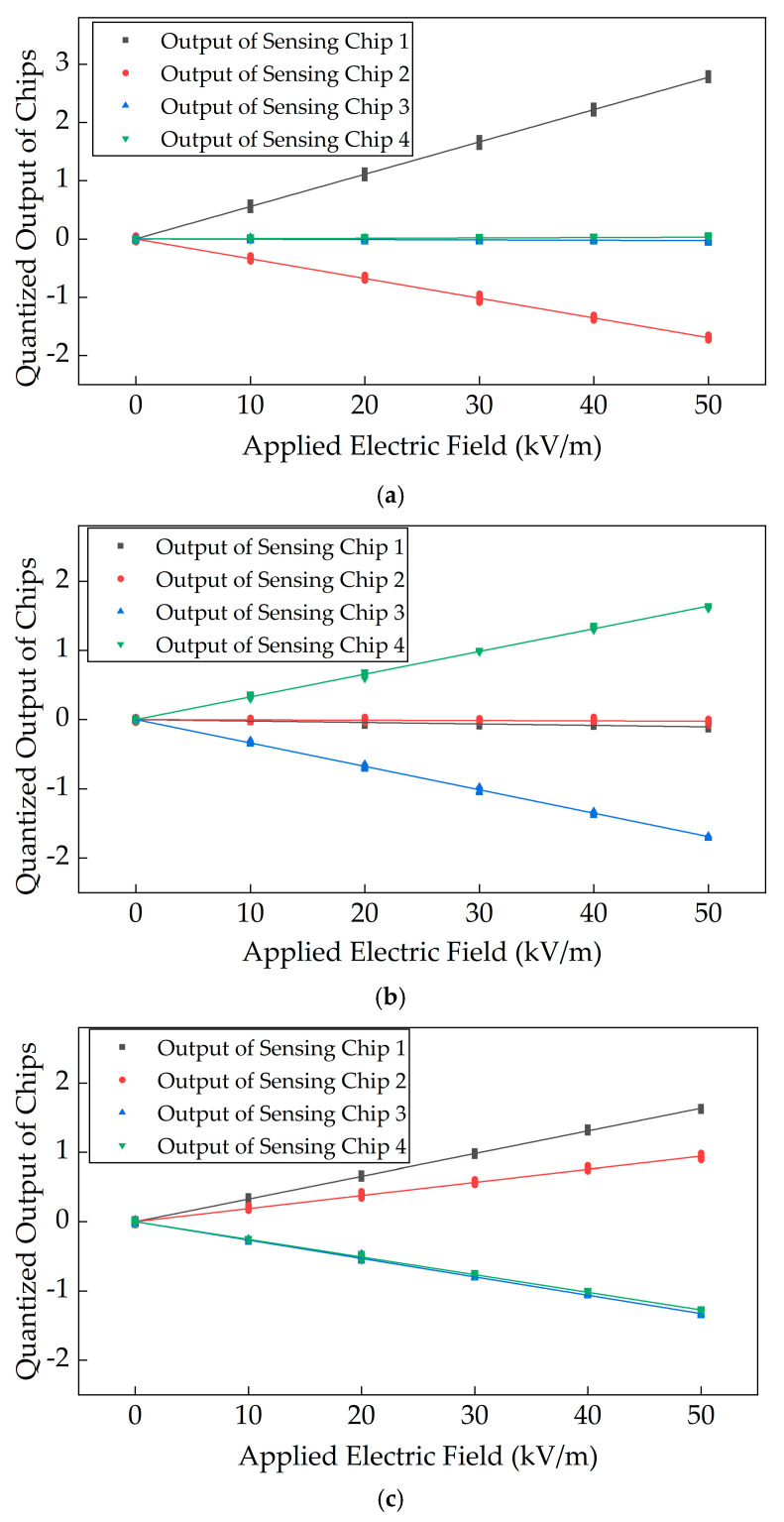
Quantization output curves of the sensing chips during three-axis calibration. (**a**) Calibration curve of the sensing chips under the electric field parallel to the *X*-axis direction; (**b**) calibration curve of the sensing chips under the electric field parallel to the *Y*-axis direction; (**c**) calibration curve of the sensing chips under the electric field parallel to the *Z*-axis direction.

**Figure 9 micromachines-16-00381-f009:**
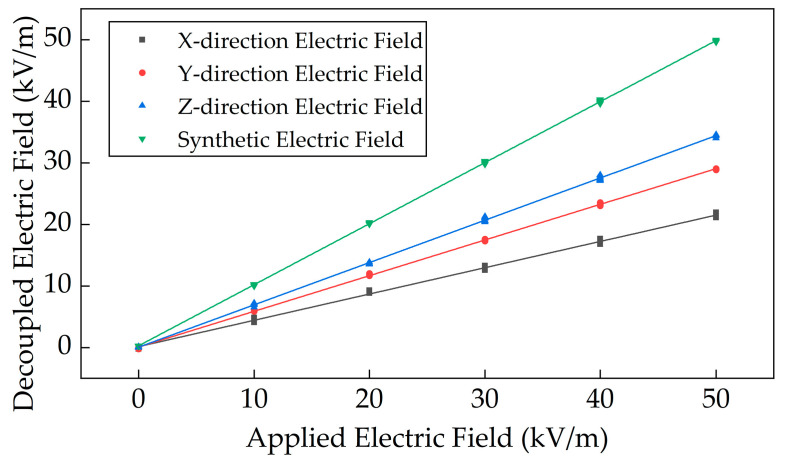
Decoupled synthetic electric field output curves under applied standard electric field.

**Table 1 micromachines-16-00381-t001:** Comparison of the synthetic electric field and relative error for two decoupling methods under different angles and applied electric fields.

Angle	Applied Electric Field (kV/m)	Chip 1 Output	Chip 2 Output	Chip 3 Output	Chip 4 Output	Sensitivity Matrix Inversion-Based Decoupling Method		Method Proposed in This Paper	
						Synthetic Electric Field (kV/m)	Relative Error (%)	Synthetic Electric Field (kV/m)	Relative Error (%)
Angle 1	30	−1.5605	0.8540	0.5017	−0.3904	29.9529	−0.1569	30.0340	0.1133
	40	−2.0517	1.2246	0.7795	−0.4241	40.1963	0.4906	39.8480	−0.3800
	50	−2.5862	1.5655	0.9748	−0.5436	50.8974	1.7947	50.4352	0.8704
Angle 2	30	−0.8947	0.5504	−0.8389	0.8597	30.0686	0.2287	30.0240	0.0799
	40	−1.2016	0.7139	−1.1234	1.1353	39.9884	−0.0290	39.9490	−0.1274
	50	−1.5097	0.8879	−1.4108	1.4097	50.0004	0.0007	49.9617	−0.0766
Angle 3	20	−0.7919	−0.0565	0.7871	−0.1510	20.1022	0.5108	20.0342	0.1709
	30	−1.1828	−0.0734	1.1926	−0.2354	30.3099	1.0331	30.2222	0.7405
Angle 4	20	0.7758	−0.5570	−0.1766	0.5028	19.3720	−3.1398	19.8506	−0.7469
Angle 5	30	−1.1879	0.6568	0.7300	−0.7096	29.8631	−0.4563	30.0415	0.1382
	40	−1.5951	0.8264	0.8914	−0.9782	39.1579	−2.1052	39.6570	−0.8575
	50	−1.9939	1.0417	1.1603	−1.1997	49.1539	−1.6921	49.6567	−0.6866
Angle 6	30	−1.3949	0.0148	1.1200	−0.0827	30.1511	0.5035	30.0829	0.2763
	40	−1.8419	−0.0434	1.4948	−0.0956	40.2161	0.5403	40.0813	0.2034
	50	−2.3049	−0.0276	1.8528	−0.1330	49.9997	−0.0006	49.8436	−0.3128
Angle 7	30	1.2808	−0.6690	−0.6763	0.6413	29.4107	−1.9642	29.6403	−1.1991
	40	1.7120	−0.9192	−0.8863	0.8711	39.5225	−1.1937	39.8358	−0.4106
	50	2.1928	−1.0571	−1.0370	1.1394	49.0831	−1.8337	49.8662	−0.2676

**Table 2 micromachines-16-00381-t002:** Relative error between the calculated and theoretical electric field values obtained by different methods.

Error (%)	Sensitivity Matrix Inversion-Based Decoupling Method	Genetic Algorithm-Based Decoupling Method [20]	Method Proposed in This Paper
Maximum Relative Error	3.14	1.90	1.20
Average Relative Error	0.98	0.59	0.43

## Data Availability

The original contributions presented in this study are included in the article. Further inquiries can be directed to the corresponding author.

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
