# Peer review of "Spatial Decoupling Method for a Novel Dual-Orthogonal Induction MEMS Three-Dimensional Electric Field Sensor"

_micromachines, 2025, doi:10.3390/mi16040381_

Round 1
Reviewer 1 Report
Comments and Suggestions for Authors
The manuscript proposes a novel MEMS 3D electric field sensor with a dual orthogonal induction structure to suppress coupling interference. It introduces a novel spatial decoupling calibration method to improve measurement accuracy. Experimental results show that the proposed method has a good linearity and is superior to traditional decoupling method in measurement errors. The paper is well-structured, with a clear research methodology and comprehensive experimental validation. I suggest the manuscript is suitable to publish after addressing the following comments.
1. The introduction should be supplemented with more recent literature on decoupling calibration methods to enhance the timeliness of the research.
2. The data analysis in Table 1 is relatively brief. It is recommended to further analyze and discuss the two decoupling methods to highlight the advantages of the method proposed in this paper.
3. There is limited discussion on the application scenarios of the sensor, as well as the challenges and limitations faced in practical applications. It is recommended to increase the analysis of the potential challenges and improvement directions in practical applications.
Comments on the Quality of English LanguageNone
Author Response
Comments 1: The introduction should be supplemented with more recent literature on decoupling calibration methods to enhance the timeliness of the research.
Response 1: Thank you for pointing this out. We agree with this comment. Therefore, we have supplemented the introduction with more recent literature on decoupling calibration methods to enhance the timeliness of the research. Specifically, we added references to recent studies on the application of neural network techniques in decoupling methods and supplemented the discussion on the latest development. The revisions can be found in the Introduction section on page 2, paragraph 2, lines 96–109.
Comments 2: The data analysis in Table 1 is relatively brief. It is recommended to further analyze and discuss the two decoupling methods to highlight the advantages of the method proposed in this paper.
Response 2: Thank you for pointing this out. We agree with this comment. Therefore, we have further analyzed and discussed the performance of the proposed method and the traditional method in all 18 tests, highlighting the advantages of our method. Specifically, we have expanded the discussion on the relative errors observed in different test conditions, emphasizing the improved accuracy and robustness of the proposed decoupling approach. The revised content can be found on page 14, paragraph 4, lines 405–413.
Comments 3: There is limited discussion on the application scenarios of the sensor, as well as the challenges and limitations faced in practical applications. It is recommended to increase the analysis of the potential challenges and improvement directions in practical applications.
Response 3: Thank you for pointing this out. We agree with this comment. Therefore, we have added a discussion on the potential challenges in real-world applications, including long-term stability and environmental adaptability, along with possible solutions. Specifically, we have elaborated on factors such as material fatigue, MEMS component degradation, and the influence of temperature and humidity, while also suggesting future research directions such as temperature compensation algorithms and improved decoupling techniques. The revised content can be found on page 16, paragraph 3, lines 461–471.
Reviewer 2 Report
Comments and Suggestions for Authors
The manuscript proposes a novel MEMS-based 3D electric field sensor with a dual orthogonal induction structure to minimize coupling interference. It introduces a spatial decoupling calibration model based on the Cramér-Rao Lower Bound (CRLB) to enhance measurement accuracy. Experimental results demonstrate significant improvements in linearity and reduced relative error compared to traditional methods. The most powerful points are: The proposed dual orthogonal induction structure and the use of CRLB for decoupling calibration appear innovative. The methodology is comprehensive, with clear descriptions of sensor design, calibration, and testing. Also, the experimental results effectively validate the proposed methods, showing a substantial reduction in measurement error. Finally, the manuscript is well-organized, with a logical flow from the introduction to the results and discussion. I see the manuscript is suitable to publish in “Micromachines” after the authors fixed these points;
1- While the manuscript provides a background, it lacks a detailed comparison with state-of-the-art techniques beyond the traditional methods mentioned. I suggest the authors include a more comprehensive comparison with recent advancements in MEMS electric field sensors and decoupling methods.
2- The mathematical derivations, particularly for the CRLB and decoupling matrix, are dense and could benefit from additional explanation. I recommend the authors provide more step-by-step explanations or a supplementary section to aid readers less familiar with these techniques.
3- Some figures and tables, especially calibration curves and decoupling matrices, lack sufficient captions and explanations. The authors should enhance captions to describe the significance of the data presented clearly.
4- The discussion on limitations and potential improvements is minimal. I recommend authors expand this section to address potential challenges in real-world applications.
Author Response
Comments 1: While the manuscript provides a background, it lacks a detailed comparison with state-of-the-art techniques beyond the traditional methods mentioned. I suggest the authors include a more comprehensive comparison with recent advancements in MEMS electric field sensors and decoupling methods.
Response 1: Thank you for pointing this out. We agree with this comment. Therefore, we have supplemented the introduction with more recent literature on decoupling calibration methods to enhance the timeliness of the research. Specifically, we added references to recent studies on the application of neural network techniques in decoupling methods and supplemented the discussion on the latest development. The revisions can be found in the Introduction section on page 2, paragraph 2, lines 96–109.
Comments 2: The mathematical derivations, particularly for the CRLB and decoupling matrix, are dense and could benefit from additional explanation. I recommend the authors provide more step-by-step explanations or a supplementary section to aid readers less familiar with these techniques.
Response 2: Thank you for your valuable suggestion. We agree with this comment. Therefore, we have added a detailed explanation of the CRLB fundamentals to aid readers in understanding its theoretical basis. Specifically, we have introduced the CRLB as a theoretical lower bound for estimation accuracy, explained its significance in evaluating estimator performance, and clarified its role in the decoupling model used in this study. The revised content can be found on page 7, paragraph 5-7, lines 236-248.
Comments 3: Some figures and tables, especially calibration curves and decoupling matrices, lack sufficient captions and explanations. The authors should enhance captions to describe the significance of the data presented clearly.
Response 3: Thank you for your suggestion. We agree with this comment. Therefore, we have enhanced the captions of Figures 7 and 8 to provide a clearer explanation of the significance of the presented data. These revisions help clarify the role of the calibration curves and decoupling matrices in our study. The updated captions can be found in page 13 and 14, Figure 7 and Figure 8, lines 377-380 and 397.
Comments 4: The discussion on limitations and potential improvements is minimal. I recommend authors expand this section to address potential challenges in real-world applications.
Response 4: Thank you for pointing this out. We agree with this comment. Therefore, we have added a discussion on the potential challenges in real-world applications, including long-term stability and environmental adaptability, along with possible solutions. Specifically, we have elaborated on factors such as material fatigue, MEMS component degradation, and the influence of temperature and humidity, while also suggesting future research directions such as temperature compensation algorithms and improved decoupling techniques. The revised content can be found on page 16, paragraph 3, lines 461–471.
Reviewer 3 Report
Comments and Suggestions for Authors
This manuscript reports an interesting MEMS 3D electric field sensor with a dual orthogonal induction structure. In addition, this manuscript incorporates the spatial decoupling method for this sensor. Experimental tests were included to measure the efficiency of the proposed method. This method suppressed coupling interference of the MEMS electric field sensor, improving the accuracy of this sensor. The proposed method can reduce the 3D coupling interference of electric field sensors. The manuscript is well organized and written. Only, the following comments must be addressed.
1.- The introduction should include more recent references (e.g., between 2024 and 2025).
2.- The resolution of Figure 1, 2, 3, 4, 6, 7, and 8 should be improved.
3.- The authors should improve the description of the design of the proposed MEMS sensor, regarding the materials and dimensions of main components.
4.- The description of the experimental equipment used in Figure 5 should be enhanced.
5.- The authors could consider the discussion of the challenges or limitations of the proposed method.
6.- The authors should include more discussions of the results reported in the Tables 1 and 2.
7.- What are the future research directions?
Author Response
Comments 1: The introduction should include more recent references (e.g., between 2024 and 2025).
Response 1: Thank you for pointing this out. We agree with this comment. Therefore, we have supplemented the introduction with more recent literature on decoupling calibration methods to enhance the timeliness of the research. Specifically, we added references to recent studies on the application of neural network techniques in decoupling methods and supplemented the discussion on the latest development. The revisions can be found in the Introduction section on page 2, paragraph 2, lines 96–109.
Comments 2: The resolution of Figure 1, 2, 3, 4, 6, 7, and 8 should be improved.
Response 2: Thank you for your suggestion. We agree with this comment. However, due to the large file size of the images, the Word document became difficult to upload. Therefore, we have compressed the images for the manuscript, but we have included a ZIP file containing all high-resolution images as an attachment.
Comments 3: The authors should improve the description of the design of the proposed MEMS sensor, regarding the materials and dimensions of main components.
Response 3: Thank you for your suggestion. We agree with this comment. We have added detailed descriptions of the materials and dimensions of the MEMS sensor housing and sensing electrodes. The revised content can be found on page 10, paragraph 2, lines 311–315.
Comments 4: The description of the experimental equipment used in Figure 5 should be enhanced.
Response 4: Thank you for your suggestion. We agree and have added details on the parallel electrode plates and high-voltage DC power supply to clarify the experimental setup. The revised content can be found on page, paragraph 1, lines 331–337.
Comments 5: The authors could consider the discussion of the challenges or limitations of the proposed method.
Response 5: Thank you for pointing this out. We agree with this comment. Therefore, we have added a discussion on the potential challenges in real-world applications, including long-term stability and environmental adaptability. Specifically, we have elaborated on factors such as material fatigue, MEMS component degradation, and the influence of temperature and humidity. The revised content can be found in page 16, paragraph 3, lines 461–466.
Comments 6: The authors should include more discussions of the results reported in the Tables 1 and 2.
Response 6: Thank you for your suggestion. We agree and have expanded the discussion in Tables 1 and 2. In Table 1, we provide a more detailed analysis of the 18 test cases, emphasizing the advantages of our method in accuracy and robustness. In Table 2, we compare our method with traditional and state-of-the-art decoupling methods, highlighting improvements in maximum and average relative errors. The revisions can be found on page 15 and 16, lines 405-413 and 417-435.
Comments 7: What are the future research directions?
Response 7: Thank you for pointing this out. We agree with this comment. Therefore, we have added a discussion on the potential challenges in real-world applications, and future research directions such as temperature compensation algorithms and improved decoupling techniques. The revised content can be found in page 16, paragraph 3, lines 466–471.